# Isolation and long-term expansion of murine epidermal stem-like cells

Jingjing Wang, Maureen Mongan, Xiang Zhang, Ying Xia*

Department of Environmental and Public Health Sciences, College of Medicine, University of Cincinnati, Cincinnati, Ohio, United States of America

* ying.xia@uc.edu

**Data Availability Statement:** The authors confirm that the data supporting the findings of this study are available within the article and its supplementary materials. The RNA-seq raw data can be accessed at GEO (GSE167437).

## Abstract

Epidermis is the most outer layer of the skin and a physical barrier protecting the internal tissues from mechanical and environmental insults. The basal keratinocytes, which, through proliferation and differentiation, supply diverse cell types for epidermal homeostasis and injury repair. Sustainable culture of murine keratinocyte, however, is a major obstacle. Here we developed murine keratinocyte lines using low-$Ca^{2+}$ (0.06 mM) keratinocyte serum-free medium (KSFM-$Ca^{2+}$) without feeder cells. Cells derived in this condition could be subcultured for >70 passages. They displayed basal epithelial cell morphology and expressed keratin (Krt) 14, but lacked the epithelial-characteristic intercellular junctions. Moreover, these cells could be adapted to grow in the Defined-KSFM (DKSFM) media containing 0.15 mM $Ca^{2+}$, and the adapted cells established tight- and adherens-junctions and exhibited increased *Krt1/10* expression while retained subculture capacity. Global gene expression studies showed cells derived in KSFM-$Ca^{2+}$ media had enriched stem/proliferation markers and cells adapted in DKSFM media had epithelial progenitor signatures. Correspondingly, KSFM-$Ca^{2+}$-derived cells exhibited a remarkable capacity of clonal expansion, whereas DKSFM-adapted cells could differentiate to suprabasal epithelial cell types in 3-dimentional (3D) organoids. The generation of stem-like murine keratinocyte lines and the conversion of these cells to epithelial progenitors capable of terminal differentiation provide the critically needed resources for skin research.

## Introduction

Epidermis renews and regenerates continuously throughout the lifetime of an organism. The keratinocyte, located at the basal layer, is the primary cell type in the epidermis that divides and serves as a source for epidermal maintenance and regeneration through self-renew and periodical differentiation. Once differentiated, the cells move outwards in a process known as basal-to-suprabasal transition to replace the cells at outer layer that are sloughed off into the environment. Differentiation is associated with up-regulation of signature genes, such as *Krt1*, *Krt10*, *Involucrin* and *Loricrin*. These gene products form the intermediate filament network that tightens cell junctions at the epidermal surface to enable protection against the hostile environment.

**Funding:** Work described here is supported in part by National Institute of Health grants RO1EY15227 (YX), RO1HD098106 (YX) and P30ES006096 supported pilot grant. The funders had no role in study design, data collection and analysis, decision to publish, or preparation of the manuscript.

**Competing interests:** The authors have declared that no competing interests exist.

The epidermal progenitors/stem cells sustain the basal keratinocyte repertoire and play key roles in epidermal renewal, maintenance, and presumably wound repair and neoplasia. *In vivo* cell kinetic tracing studies in murine epidermis suggest the existence of diverse subgroups of stem cells and progenitors with unique characteristics [1]. A small subgroup, known as epithelial stem cells, is characterized as relative quiescent and slow-cycling, but having a great proliferative potential and an unlimited capacity of self-renewal [2, 3]. A relatively large subgroup is the transit amplifying (TA) cells. These cells have rapid but limited proliferative capacity and eventually depart from the basal to the suprabasal layer, accompanied with terminal differentiation within 4–5 days. There are also subgroups bearing some stem and some TA features, and subgroups that differentiate in the early stages of keratinization while still capable of proliferation. The epidermis, therefore, possesses a collection of highly heterogeneous populations of cells with diverse molecular and functional traits. The identification and isolation of stem cells and progenitors from the skin have remained challenging.

Mice have emerged as valuable models to study the genetic regulation of skin biology and the mechanisms of skin diseases; however, the murine keratinocytes are difficult to amplify *in vitro*. These cells have so far been limited mostly to primary culture, in contrast to their human counterparts that are easy to subculture [4–6]. This limitation precludes obtaining a sustainable cell resource to study epithelial abnormalities. Here we tested commercial growth media to culture murine keratinocytes. We found that the keratinocyte serum-free medium without calcium (KSFM-$Ca^{2+}$), supposedly depleted of calcium chloride, contained low but detectable levels of $Ca^{2+}$ at 0.06 mM. While this medium was not widely used for keratinocyte culture, it surprisingly allowed survival of a few cells that grew into clones, and the clones continuously proliferated that ultimately grew to confluence. Cells derived under this condition displayed epithelial stem cell characteristics and could be subcultured for >70 passages. Moreover, these cells could adapt to grow in Defined K-SFM (DKSFM) media containing 0.15 mM $Ca^{2+}$ and the adapted cells retained subculture capacity. Distinct from the parental cells, the DKSFM-adapted cells exhibited epithelial progenitor signatures, could form organoids in 3D culture and be induced to terminally differentiate. The *in vitro* culture conditions described here enable acquiring murine epithelial stem cell-like and progenitor-like lines for biomedical research.

## Materials and methods

### Chemicals, reagents and antibodies

Soybean Trypsin Inhibitor (17075–029) was from Gibco. $CaCl_2$ solution (C-34006) was from PromoCell; Collagen IV (354233) was from BD Biosciences Discovery; calcium concentration kit (MAK022-1KT) and Hoechst 33342 (B2261) were from Sigma. All antibodies, cell culture media and reagents are listed in S1 and S2 Tables in S2 File.

### Experimental animals and cell culture

C57/BL6 newborn pups were used for the isolation of keratinocytes following procedures described [5]. Briefly, pups were sacrificed by decapitation. Skins were collected and placed in 0.25% trypsin with dermis side down. After incubation overnight at 4˚C, the epidermal sheets separated from the dermis were collected in a tube containing 5 ml serum-free media with trypsin inhibitors (1 mg/ml). After vigorous shaking, the epidermal sheets were removed and the cells remaining in the media were collected by centrifugation. The cell pellets were resuspended in the culture medium of choice and seeded at $1x10^5$ cells/$cm^2$ on culture plates pre-coated with collagen IV (ColIV, 7 μg/ml). All mice experiments were in adherence to a

protocol approved by the University of Cincinnati Animal Care and Use Committee and followed the National Institutes of Health guide for the care and use of Laboratory animals.

Cells were grown in a humidified $CO_2$ incubator at 37°C and refed every 3 days. Subculture was carried out when cells reached 80–90% confluence. After rinsing with PBS, the cells were submerged in 0.05% EDTA in PBS for 10 min at 37°C, followed by digestion twice with TrypLE at 37°C for 10 min. The detached cells were collected in growth media containing trypsin inhibitor (1 mg/ml) and pelleted by centrifugation. The cells were resuspended in the medium of choice, plated at 1–2.5 x $10^4$/$cm^2$ on ColIV-coated tissue culture dishes and incubated at 37°C in a humidified $CO_2$ chamber. Cell pellets could also be resuspended in culture medium containing 10% DMSO and stored in liquid $N_2$, and recovered later for culture.

Mouse embryonic stem cell (mESC) C-2 line was maintained in culture as described before [7]. Mouse embryonic fibroblast cell line STO (ATCC CRL-1503) and primary mouse embryonic fibroblasts (MEFs) were grown in DMEM supplemented with 10% fetal bovine serum, 2 mM glutamine, 1% nonessential amino acids, 1 mM sodium pyruvate, 100 U/ml penicillin and 100 μg/ml streptomycin. Cells were grown in a humidified $CO_2$ (5%) incubator at 37°C and passaged when reached 80% confluence.

## Calcium concentration measurement

The calcium concentrations in the growth media were determined using calcium concentration kit following the manufacturer's protocol. A serial dilution of the CaCl2 solution was used as standards.

## Telomere length measurement

The telomere length in genomic DNA was assessed with real-time qPCR [8]. Briefly, telomere repeat was determined by the comparative Ct method—the Ct values of telomere repeats were normalized with the Ct values of a single-copy gene, i.e. 36B4 (acidic ribosomal phosphoprotein P0). Each sample was examined in triplicates and the telomere length was calculated based on the difference in the ΔCt values between telomere repeats and 36B4 at 2^(-ΔΔCt).

## Karyotyping

Primary keratinocytes in DKSFM medium (P0) and KSFM-$Ca^{2+}$ cells at passage 5, 25 and 30 were subjected to chromosomal analysis using Giemsa staining (Sigma) [9]. The ESCs and MEFs were used as controls. At least 12 metaphase cells under each condition were examined.

## Colony-forming efficiency (CFE) assays

Cell suspension was plated at 300 cells/$cm^2$ and medium was refreshed every 3 days. At 14 days in culture, the medium was removed, cells were fixed with 4% paraformaldehyde (PFA) for 15 min, washed twice with PBS, and stained with 1% crystal violet at room temperature for 30 min. The plates were rinsed with ample water, dried and photographed. Colonies were counted and CFE was calculated using a formula: (# colonies/#total cells plated) * 100.

## Three-dimension (3D) organoids

The 3D culture was carried out following a protocol previously described [10, 11]. In short, single cell suspension was mixed with Basement Membrane Extract (BME) at 2x $10^5$/ml and set on ice; 10 ul cell/BME mixture was placed on tissue culture dishes as droplets, and the droplets were allowed to solidify at 37°C for 2 min before being covered by pre-warmed growth media with or without $Ca^{2+}$ supplements. The organoids were collected on day 14 in culture.

## Histological and immunofluorescence staining

Cultured cells grown on Col IV-coated 6 mm glass coverslips and collected organoids were fixed with 4% paraformaldehyde at 4˚C for 30 min. The cells were permeabilized using PBS with 0.2% Triton and subjected to immunofluorescent staining. The organoids were dehydrated, embedded in paraffin wax and sectioned; the sections were subjected to Hematoxylin and Eosin (H&E), immunohistochemistry and immunofluorescent staining [12]. For immunostaining, primary antibodies were diluted at 1:100 and secondary antibodies and nucleus staining reagents were dilute at 1:400. Images were visualized and captured using a Zeiss Axio microscope.

## RNA isolation, reverse transcription and quantitative polymerase chain reaction (qPCR)

Total RNA was isolated using PureLink RNA Mini Kit (12183025, Invitrogen); reverse transcription was performed using SuperScript IV reverse transcriptase (18090010, Invitrogen); qPCR was carried out with an Agilent Technologies Stratagene Mx3000P PCR machine using PowerUp SYBR Green Master Mix (4367659, Applied Biosystems) as the detection format. The PCR reactions ran for 40 cycles under the appropriate parameters for each pair of primers and fluorescence values were used to construct the amplification curve. The results were normalized using *Gapdh* and $\Delta\Delta Ct$ were used to calculate fold change. Data represent results of triplicates in 2 or more experiments. The sequences of PCR primers are listed in S3 Table in S2 File.

## Global gene expression and data analyses

Directional poly A RNA-seq was performed by the Genomics, Epigenomics and Sequencing Core at the University of Cincinnati using established protocols as previously mentioned [13]. Details of RNA-seq and differential gene expression analyses are included in the S1 File. The RNA-seq data were deposited at Gene Expression Omnibus (GEO) publicly accessible (GSE167437). Functional enrichment analyses were conducted using Metascape as previously described [14].

## Western blot analyses

Cells were lysed in RIPA buffer (150 mM NaCl, 1% Nonidet P-40, 0.5% sodium deoxycholate, 0.1% SDS, 50 mM Tris, pH 7.4); 50 μg lysates were subjected to SDS–PAGE. Proteins on the SDS-PAGE gels were transferred onto nitrocellulose membranes and probed with antibodies as described previously [15]. Anti-PAI-1 and anti-pIκBα antibodies were diluted 1:1000. Anti-β-Actin diluted at 1:5000 was used as loading control. Anti-mouse-HRP and anti-rabbit-HRP antibodies were diluted in 1:2000.

## Statistical analyses

Means and standard deviations were calculated based on at least three independent experiments, and analyzed using student's two-tailed *t*-test. *$p<0.05$, **$p<0.01$ and ***$p<0.001$ were considered statistically significant.

# Results

## Growth and amplification of murine keratinocytes

Because calcium ion is crucial for keratinocyte proliferation and differentiation [16], we measured $Ca^{2+}$ levels in three commercial media: (i), KSFM without $Ca^{2+}$ (KSFM-$Ca^{2+}$), (ii),

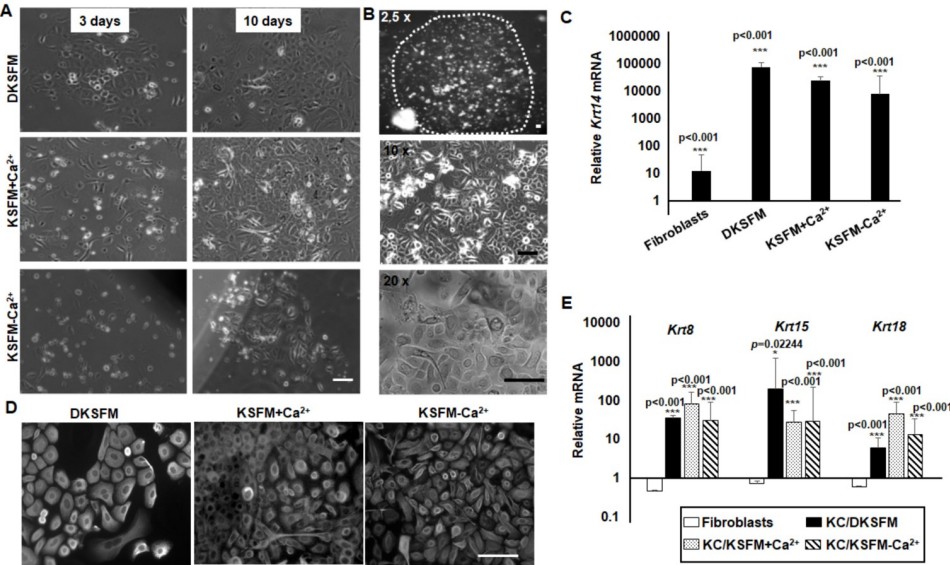

**Fig 1. Mouse keratinocyte culture in different conditions.** Bright-field microscopic images of (A) cells grown for 3 or 10 days in different media, and (B) colonies formed when cells were grown in KSFM-Ca²⁺ medium for an extended period of time. The relative expression of (C) the basal keratinocyte gene *Krt14* and (E) epithelial marker genes, i.e. *Krt8*, *Krt15* and *Krt18*, in cultured keratinocytes and fibroblasts compared to those in mESCs. (D) Cells were subjected to immunofluorescence staining with anti-KRT14; photographs were taken under fluorescent microscope. Results represent data of at least three duplicate experiments +/-SD. The scale bars in microscope image correspond to 100 μm.

KSFM with Ca²⁺ (KSFM+Ca²⁺), and (iii), defined KSFM (DKSFM). While the KSFM+Ca²⁺ and DKSFM contained ~ 0.15 mM calcium, the KSFM-Ca²⁺ had trace amount (0.06 mM) that were ~60% lower than the others (S1 Fig in S3 File).

The three media were used to culture murine keratinocytes freshly isolated from newborn pups. Cells attached well on ColIV-coated plates, proliferated and became confluent in 1–2 weeks in DKSFM and KSFM+Ca²⁺ media (Fig 1A). In contrast, cell attachment and growth were relatively inefficient in KSFM-Ca²⁺ medium. A few attached cells that appeared to be small in size grew into visible clusters by 1–2 weeks. Some of the clusters, interestingly, continued to grow into large colonies in 1–2 months, and the cells ultimately populated the entire plate in a few months (Fig 1B). Cells in all growth media displayed epithelial morphologies with regular and polygonal shapes (Fig 1A and 1B). They expressed robustly the basal keratinocyte gene *Krt14* and had Keratin (Krt) 14 proteins detectable in cytosol and perinuclei (Fig 1C and 1D). Several other keratinocyte markers, such as *Krt15*, *Krt 8* and *Krt 18*, were also abundantly expressed in these cells, in contrast to the minimal expression of these genes in mouse embryonic stem cells (mESCs) and fibroblasts (Fig 1E).

Cells grown in either KSFM+Ca²⁺ or DKSFM media could not be amplified by passaging, consistent with the recognized challenges of mouse keratinocyte subculture [17–20]. Cells grown in KSFM-Ca²⁺ medium, on the other hand, could be extensively subcultured for at least 70 times, resulting in a >10³⁰-fold increase in cell number. The longest culture was carried for up to 20 months. Several keratinocyte lines derived in KSFM-Ca²⁺ exhibited similar growth characteristics: the doubling time was initially 5.2 +/- 1.2 days, but decreased gradually with increased passage number and ultimately reached a steady doubling of ~ 0.7 days (Fig 2A–2C).

The primary somatic cells have limited proliferative capacity and enter senescence after a finite number of *in vitro* cell divisions. Senescence is associated with increased expression of

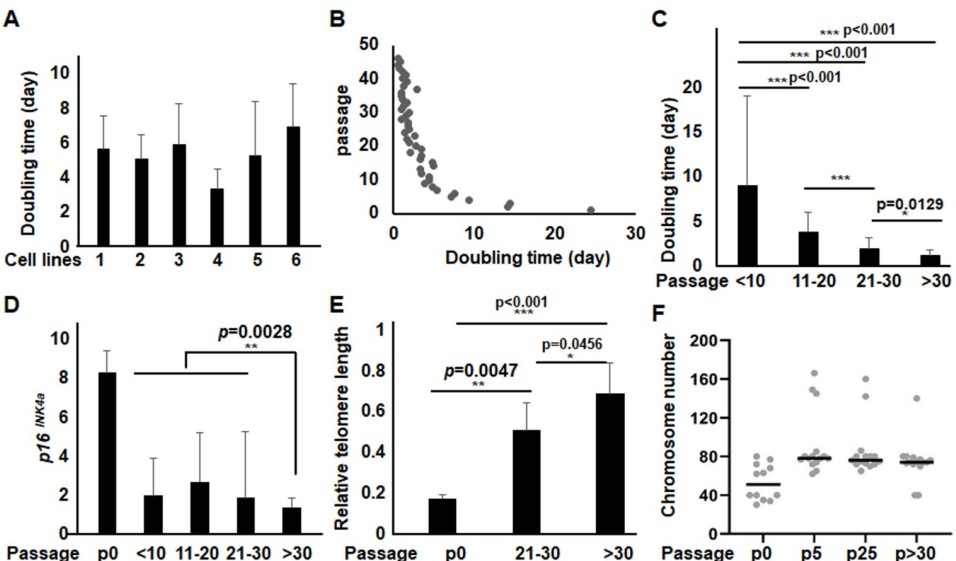

**Fig 2. The growth properties of long-term cultured keratinocytes derived in KSFM-Ca$^{2+}$ medium.** The average doubling time of representative cell lines at (A) the early passages, and (B) at different passages was calculated based on the cell numbers plated, harvested and days in culture. (C) Comparison of the average doubling time +/-SD of cells at <10, 11–20, 21–30 and >30 passages. Cells of medium and high passages grew significantly faster than those of low passages. The primary murine keratinocytes grown in DK-SFM (p0) and the KSFM-Ca$^{2+}$ cells at different passages were examined for (D) p16$^{INK4a}$ mRNA, (E) The relative telomere length, in comparison to levels in mESCs, set as 1, and (F) chromosome numbers and results represent data from 12–15 metaphase cells examined.

p16$^{INK4a}$ and shortening of telomeres along cell passages, as seen in the primary mouse embryonic fibroblasts (MEFs) [21, 22] (S1B and S1C Fig in S3 File). The KSFM-Ca$^{2+}$ cell lines had reduced *p16$^{INK4a}$* expression and increased telomere length compared with the primary keratinocytes (Fig 2D and 2E). When reaching a steady state growth at > 30 passages, the cells had a slightly further *p16$^{INK4a}$* decrease and telomere length increase. Karyotyping showed that in contrast to the mESC and MEFs, which displayed normal chromosomes (i.e. n = 40), the primary murine keratinocytes (P0) derived in DKSFM were mostly tetraploid or nearly tetraploid consistent with observations made by others [23] (Fig 2F and S1D Fig in S3 File). The long-term cultured KSFM-Ca$^{2+}$ cell lines also displayed tetraploid and hyperdiploid, and cells at early (p5) and late (>p30) passages had similar chromosome numbers (Fig 2F). Taken together, these data suggested that the KSFM-Ca$^{2+}$ medium selects a population of keratinocytes that were genetically stable with long-term proliferative potential.

After recovered from liquid nitrogen storage, the cells retained the growth and subculture capacity. These results suggest that while the KSFM+Ca$^{2+}$ and DKSFM media support primary keratinocyte proliferation, the KSFM-Ca$^{2+}$ medium enables the survival of keratinocytes with long-term proliferative output.

## Formation of keratinocyte intercellular junctions

The epithelial cells join to each other through a set of intercellular junctions that are crucial for the mechanical and functional integrity of the epidermis [24]. As part of the epithelial junction complex, the adherens junctions provide adhesiveness between cells, while the tight junctions form a paracellular diffusion barrier that regulates epithelial permeability [25, 26]. The E-cadherin (E-Cad), a component of the transmembrane adhesion complex, was detected

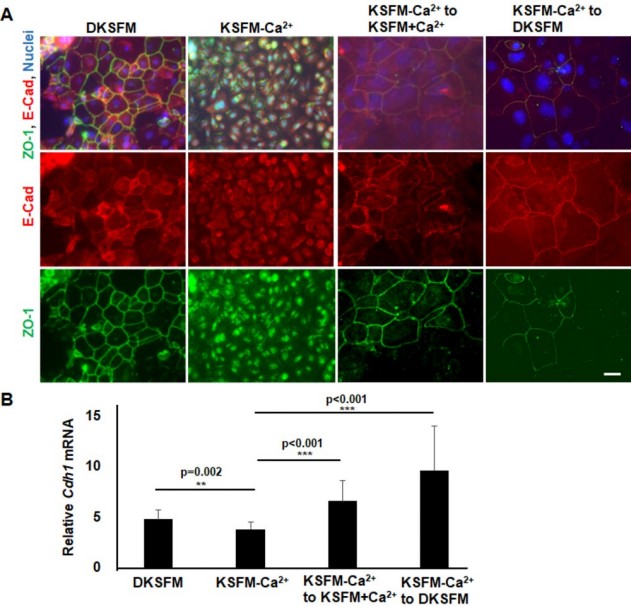

**Fig 3. Formation of cell-cell junctions depends on extracellular calcium.** (A) Primary keratinocytes grown in DKSFM and cell lines grown in KSFM-Ca$^{2+}$ with or without switching to DKSFM and KSFM+Ca$^{2+}$ media for 48 h were examined (A) by immunofluorescence staining using anti-E-Cad and anti-ZO-1, and (B) by qPCR for mRNA of E-Cadherin (*Cdh1*) compared to the level in mESCs as 1. Data represent at least triplet samples +/-SD. Scale bar represents 20 μm.

predominantly on the plasma membrane at cell-cell junctions in primary keratinocytes grown in DKSFM (Fig 3A); however, it was found in the perinuclear and dispersed in the cytoplasm in cells derived in KSFM-Ca$^{2}$. Similarly, the tight junction protein ZO-1 located at membrane and intercellular contacts of cells grown in DKSFM appeared in the nucleus of cells derived in KSFM-Ca$^{2}$.

To test if it was the low extracellular calcium that prevented the formation of intercellular junctions in KSFM-Ca$^{2}$-derived cells, we placed the cells in either KSFM+Ca$^{2+}$ or DKSFM media, a change that resulted in an increase of extracellular calcium from 0.06 mM to 0.15 mM. Both the adherens and the tight junctions were restored in the KSFM-Ca$^{2+}$-derived cells after growing for 2 days in media containing 0.15 mM Ca$^{2+}$, corresponding to a significant up-regulation of E-Cad expression (Fig 3A and 3B). Thus, increasing extracellular calcium could restore the cell-cell junctions in the KSFM-Ca$^{2}$-derived cells.

When transferred to DKSFM media, the KSFM-Ca$^{2+}$-derived cells increased in size, followed by a massive cell loss after one week. Nonetheless, a small percentage of the cells was able to survive and grow, and eventually repopulated to reach confluence (Fig 4A). These DKSFM-adapted cells retained subculture capacity, could be passaged 34 times so far, and storage in liquid nitrogen. Moreover, they exhibited decreased doubling time along with increased passage numbers, resembling the growth features of KSFM-Ca$^{2}$-derived cell lines (Fig 4B). Distinct from the parental cells, however, the DKSFM-adapted cells formed tight- and adherens-junctions accompanied with increased E-Cad expression (Fig 4C and 4D). These observations suggest that the atypical KSFM-Ca$^{2+}$-derived keratinocytes (so forth called KSFM-Ca$^{2+}$ cells) could be converted in DKSFM to keratinocytes that formed intercellular junctions (so forth called DKSFM cells), and that both cell lines have extensive subculture capacities.

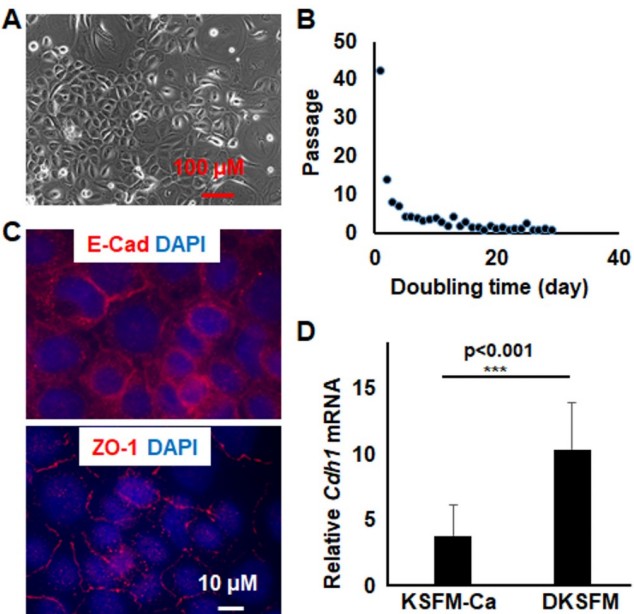

**Fig 4. Characterization of epithelial cells adapted in DKSFM media.** Cells adapted to grow in DKSFM were (A) examined with microscope and bright-field images captured, (B) examined for growth rate through the calculation of doubling time at different passages, (C) subjected to immunofluorescence staining using antibodies for E-Cad and ZO-1, and images were captured with Zeiss fluorescent microscope, and (D) measured for E-Cad (*Cdh1*) expression by qPCR and expression was compared to that in mESCs as 1. Results represent duplicate data of at least 3 biological samples.

## Gene expression signatures of the epithelial cell lines

To understand the molecular features of the cell lines, we examined the global gene expression signatures using RNA-seq. In 9006 genes detected at an adjusted $p<0.01$, approximately 35% of them were differentially expressed ($> = 2$-fold) between the KSFM-Ca$^{2+}$ and DKSFM cells (Fig 5A). The differential gene expression signatures indicate that the KSFM-Ca$^{2+}$ cells have up-regulated functions in immune/inflammatory/cytokine responses and DNA replication/ mitosis/cell cycle regulation, whereas the DKSFM cells have up-regulated functions in epithelial cornification/differentiation/adhesion/morphogenesis (Fig 5B and 5C). Compared to the KSFM-Ca$^{2+}$ cells, the DKSFM cells had more mRNA transcripts of cytokeratins (*Krt*), and genes implicated in epidermal terminal differentiation, cell-cell adhesion and cell-matrix interactions (S2 Fig in S3 File), but less transcripts of genes of inflammatory/immune responses and cell cycle/proliferation (S3 Fig in S3 File).

Selective gene expression was validated by qPCR. Specifically, compared to the KSFM-Ca$^{2+}$ cells, the DKSFM cells displayed a slight increase in expression of basal keratinocyte marker *Krt14*, more elevated expression of spinous layer markers *Krt1* and *Krt10*, and a granular layer marker Loricrin (*Lor*); they also had more abundant expression of *Krt8*, a marker for simple single-layered epithelial cells [27] (Fig 6A). Also consistent with RNA-seq findings suggesting that KSFM-Ca$^{2+}$ cells had elevated inflammatory responses (S4 Table in S2 File), qPCR detected significantly increased expression of an inflammatory mediator C-C Motif Chemokine Ligand 2 (*Ccl2*) in KSFM-Ca$^{2+}$ over DKSFM cells (Fig 6B). The mRNA and proteins of the inflammation biomarker Plasminogen activator inhibitor-1 (*Pai-1*) were more abundant in KSFM-Ca$^{2+}$ but hardly detectable in DKSFM cells [28, 29] (Fig 6B and 6C). Moreover,

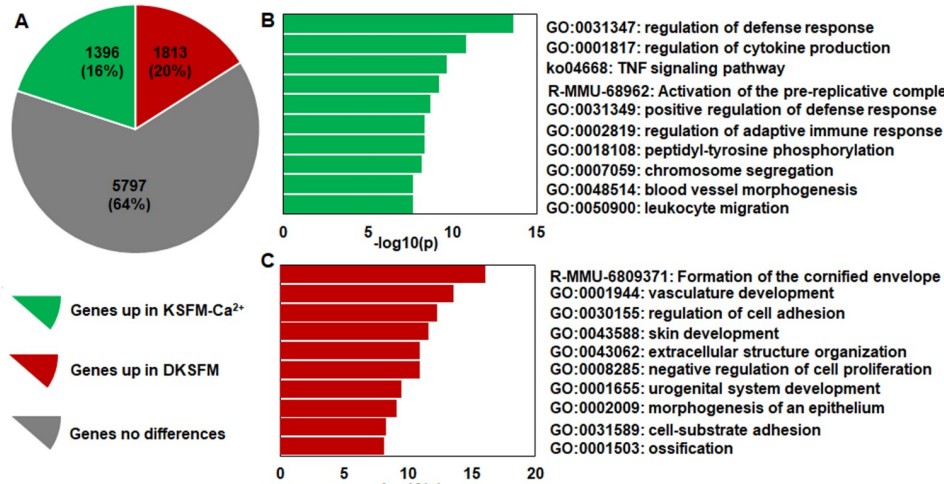

**Fig 5. Global gene expression and RNA-seq analysis.** (A) Of the significant genes detected in the KSFM-Ca$^{2+}$ and DKSFM cells, 16% were significantly up-regulated in the KSFM-Ca$^{2+}$ cells whereas 20% were significantly up-regulated in the DKSFM cells. Enrichment analysis reveals the top functions up-regulated in (B) KSFM-Ca$^{2+}$ cell and (C) DKSFM cells.

phosphorylation of IkBα, a direct indicator of the TNF pathway activation [30], was higher in KSFM-Ca$^{2+}$ than DKSFM cells.

To evaluate the differential proliferation of the KSFM-Ca$^{2+}$ and DKSFM cells, we examined the expression of *proliferating cell nuclear antigen* (*Pcna*), encoding a protein that regulates DNA replication, and *Ki67*, a DNA-binding proliferative protein [31]. The qPCR showed that *Pcna* and *Ki-67* mRNA was increased by nearly 2-fold in KSFM-Ca$^{2+}$ over DKSFM cells.

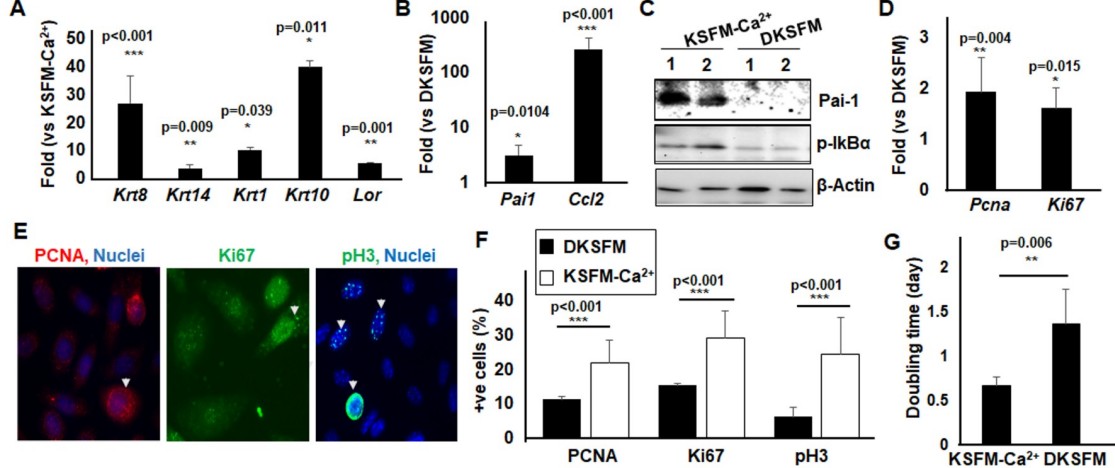

**Fig 6. Differential functional pathways in KSFM-Ca$^{2+}$ and DKSFM cells.** Differential gene expression in KSFM-Ca$^{2+}$ and DKSFM cells were validated with qPCR, on (A) epithelial differentiation markers, including *Krt1, Krt8, Krt10, Krt14 and Lor*, (B) inflammatory genes *Ccl2* and *Pai-1*, and (D) *Pcna* and *Ki67* implicated in proliferation. The expression at protein levels of (C) PAI-1 and pIκBα was examined by Western blotting and β-Actin was used as a loading control. The expression of PCNA, Ki67 and pH3 (E) was examined by immunofluorescent staining and (F) the percentage of staining positive cells were quantified based on biological triplicates. Arrowheads pointed at typical staining positive cells. (G) The doubling time of the KSFM-Ca$^{2+}$ cells and the DKSFM cells was calculated and results represent the average of >20 data sets. Statistical analyses were performed with Student's *t*-test.

Correspondingly, immunostaining detected enrichment of PCNA and Ki-67 in the nucleus of approximately 22% KSFM-Ca$^{2+}$ cells versus ~11% DKSFM cells (Fig 6D–6F). In addition, the phosphorylation of histone 3 (pH3), a highly specific proliferation marker where phosphorylation takes place only in mitosis [32], was detectable in ~20% KSFM-Ca$^{2+}$ cells but only in less than 5% of DKSFM cells. The doubling time of the KSFM-Ca$^{2+}$ cells was half that of the DKSFM cells, supporting the contention that the KSFM-Ca$^{2+}$ cells grew faster (Fig 6G). These results reveal distinct characteristics of the KSFM-Ca$^{2+}$ and DKSFM cells in differentiation, proliferation and inflammatory pathway activation.

## The keratinocyte lines display epidermal stem cell characteristics

The unlimited subculture capacities of the keratinocyte lines prompted us to ask whether these cells possessed stem cell properties. We tested the possibility by examining the RNA-seq data for expression of epidermal stem cell markers. Both the KSFM-Ca$^{2+}$ and the DKSFM lines had abundant expression of epidermal stem/progenitor markers, including *Fst*, *Tnc*, *Itga6* and *Trp63*, underscoring their stemness potential [33] (Fig 7A). However, compared to the KSFM-Ca$^{2+}$ cells, which had more abundant expression of *Cd200*, *Cd34*, *Cd44*, *Fzd-3* and *Itgb1* associated with hair follicle stem cells [33–35], the DKSFM cells had more abundant expression of *Lgr5*, *Krt15*, *Krt6b*, *Gli* and *Sca-1* that are markers of epidermal progenitors [36–38] (Fig 7A and 7B).

To test the stem cell properties, we performed colony-forming efficiency (CFE) assay, one of the most widely used experimental approaches to assess the proliferative ability of individual stem and progenitor cell within a biological sample [39]. While both KSFM-Ca$^{2+}$ and DKSFM cells were able to form colonies, the colonies' appearances were quite different. Compared to the DKSFM cells, the KSFM-Ca$^{2+}$ cells had larger holoclone-like colonies with uniform, round

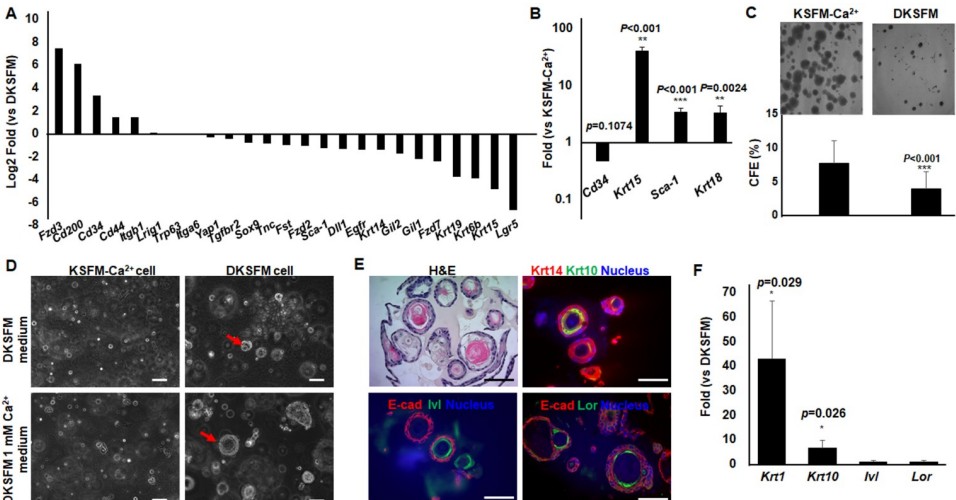

**Fig 7. Stem and differentiation characteristics of the KSFM-Ca$^{2+}$ and DKSFM cells.** (A) RNA-seq revealed common and unique stemness gene expression signatures of KSFM-Ca$^{2+}$ and DKSFM cells. (B) The expression of selective stem genes, i.e. *Cd34*, *Sca-1*, *Krt 15*, *Krt18*, were validated by qPCR. (C) Photographs of colonies formed with KSFM-Ca$^{2+}$ and DKSFM cells and the numbers of colonies were counted and CFE calculated. (D) Organoids in 3D BME culture in different culture media were photographed under microscopy in brightfields. Arrows point at the representative organoids. The DKSFM cell-derived organoids grown in DKSFM media plus 1 mM Ca$^{2+}$ were (E) subjected to histological analyses with H&E staining and immunostaining using antibodies indicated, and (F) examined by qPCR for expression of keratinocyte terminal differentiation genes, i.e. *Krt1*, *Krt10*, *Ivl*, and *Lor*. The scale bars in microscope images correspond to 100 μm. Statistical analyses were performed with Student's t-test using data derived from at least triplicate biological samples.

shapes and compact cell contents, suggesting greater stemness potential [39] (Fig 7C). Additionally, the overall colony forming efficiency (CFE) was 8% for KSFM-Ca$^{2+}$ cells in contrast to about half that value for DKSFM cells.

We further applied these cells to 3D cultures, where epithelial progenitors could form organoids under conditions that mimic the *in vivo* microanatomy [40]. We embedded the single cell suspensions of KSFM-Ca$^{2+}$ and DKSFM cells in basement membrane extract (BME) and cultured them in different media, including KSFM-Ca$^{2+}$, DKSM or DKSM plus 1 mM calcium chloride. The KSFM-Ca$^{2+}$ cells failed to form organoids in any of the growth media tested, whereas the DKSFM cells formed relatively small organoids in DKSFM media but generated large compact organoid structures in DKSFM with 1 mM Ca$^{2+}$ (Fig 7D and S4A Fig in S3 File).

The large organoids formed with the DKSFM cells were characterized on differentiation to various epithelial cell types. Immunohistochemistry of paraffin sections detected the expression of Krt14 in cells at the outer layer, and Krt10, Involucrin (Ivl) and Loricrin (Lor) in cells at the inner layers of the organoids (Fig 7E). By qPCR, we found a robust induction of *Krt1* and *Krt10* expression while a small increase in *Ivl* and *Lor* expression in the organoids (Fig 7F), in contrast to the 2D cultures where increasing extracellular calcium was unable to induce DKSFM cell terminal differentiation (S4B Fig in S3 File). Collectively, these data show that the KSFM-Ca$^{2+}$ and DKSFM cells represent unique epidermal stem/progenitor subpopulations with distinct stem cell markers, and proliferation and differentiation potentials.

## Discussion

We describe the establishment of murine epithelial cell lines under conditions free of serum and feeder cells using media containing minimal Ca$^{2+}$ (KSFM-Ca$^{2+}$). These cells express keratinocyte markers, have robust proliferation potential and can be extensively subcultured; they can also be recovered after storage in liquid N2, enabling enduring resources. Tetraploid or nearly tetraploid detected in the KSFM-Ca$^{2+}$ cells are observed also in the primary keratinocytes, suggesting chromosomal abnormality is a common feature of *in vitro* murine keratinocyte culture not resulting from extensive passaging [23]. The KSFM-Ca$^{2+}$ cell lines, on the other hand, are likely genetic stable because the early and late passaged cells have similar chromosome numbers. Cells derived in KSFM-Ca$^{2+}$ media, however, have low E-Cad expression and lack the formation of intercellular junctions, suggesting that they belong to an atypical subgroup of epidermal cells. Nonetheless, after adaption in DKSFM media containing 0.15 mM Ca$^{2+}$, the adapted DKSFM cells while retaining keratinocyte marker expression and the capacity of extensive subculture have higher E-Cad expression, tight- and adherence junction formation and can differentiate to suprabasal keratinocytes in 3D organoids. In contrast to the previously reported methods for murine keratinocyte culture, which rely on feeder cells and serum, undergo incomplete differentiation, or have limited subculture capacity [2, 5, 18, 41–43], the approaches described here promise endless cell supplies and advantageous experimental tools for skin biology research.

The proliferation capacity of the epidermis is sustained by progenitors, transient amplifying and stem cells, located in the interfollicular epidermis and the inner sheath of hair follicles [44–47]. We find the KSFM-Ca$^{2+}$ cells possess stem-like characteristics, i.e. colony formation and signature gene expression, but do not exhibit senescence features, such as $p16^{INK4a}$ increase and telomere shortening [48]. Our data further suggest that the KSFM-Ca$^{2+}$ cells are located above the DKSFM cells in the stemness hierarchy. Several pieces of evidence lend supports to this contention. First, the KSFM-Ca$^{2+}$ cells proliferate faster with shorter doubling time and more abundant expression of genes implicated in cell cycle and DNA replication,

such as *Ki67* and *Pcna*. These characteristics are consistent with those of stem cells due to abbreviated G1 phase, absent G0 phase and differentially regulated cell cycle checkpoints [49, 50]. Second, the KSFM-Ca$^{2+}$ cells lack cell-cell junctions, in agreement with the observed stem cell morphologies that prevent cell adhesion and junction formation [51]. Third, the KSFM-Ca$^{2+}$ cells have potent clonogenic potential, forming larger holoclones with smooth perimeter, in contrast to the DKSFM cells that have less and smaller colonies likely due to decreased stem and increased differentiation potentials [39]. Fourth, the KSFM-Ca$^{2+}$ cells display increased inflammatory gene signatures and activation of TNF signaling pathways, characteristics associated with epidermal stem cells [35]. In this context, the KSFM-Ca$^{2+}$ cells may represent a subtype of stem cells that possesses inflammatory memory, responsible for quick barrier restoration in response to tissue damage [52]. A key regulator of the memory is Aim2, encoding an activator of the inflammasome [53]. Aim 2 expression is up-regulated by 30-fold in KSFM-Ca$^{2+}$ versus DKSFM cells in the RNA-seq data.

Formation of self-organizing organoids with structures mimicking the native microanatomy in 3D culture is a unique feature of stem and stem-like cells [40]. We find that the DKSFM, but not KSFM-Ca$^{2+}$, cells are capable of forming 3D organoids. These organoids are composed of basal epithelial cells at the outer layer, and the keratinized inner core expressing not only Krt10, but also Involucrin and Loricrin that contribute to the cornified envelops. Recently, Boonekamp et. al. have shown that it is the interfollicular epithelial (IFE) progenitor that contributes to the 3D organoids formation [11]. Accordingly, we find that compared to the KSFM-Ca$^{2+}$, the DKSFM cells have more abundant expression of the IFE markers *Sca-1* and *Lgr5*. In addition, the DKSFM cells have enriched expression of *Krt19*, a marker of hair follicle absent in interfollicular epidermal cells [54], and increased expression of *Dll1*, a marker of interfollicular stem cells [55]. Possibly, the DKSFM cells represent a mixture of various stem, IFE progenitors and differentiating cell subtypes, capable of terminal differentiation under a suitable micro-environment.

Given that the KSFM-Ca$^{2+}$ cells arise from a few cell clusters with a relatively low cluster forming efficiency (~ 0.06%), we postulate that these cells represent a small subset of epidermal stem cells that survive under the KSFM-Ca$^{2+}$ conditions. The stem cell marker expression patterns suggest that these cells come from the hair follicles, and specifically, from the infundibular and isthmus epithelia, whose unique markers, Ccl2 and Ccl20, are expressed abundantly in the KSFM-Ca$^{2+}$ cells [56]. Despite these observations, whether the KSFM-Ca$^{2+}$ cells originate from cells residing in the upper and middle segments of the hair follicles is yet to be determined.

*In vivo*, low Ca$^{2+}$ concentrations are found in the basal epidermis, where epithelial cells proliferate to maintain homeostasis, whereas high Ca$^{2+}$ concentrations are in the suprabasal layers associated with terminal differentiation and keratinization [16]. Here we show that the KSFM-Ca$^{2+}$ media containing minimal calcium can be used to obtain murine keratinocytes with epidermal stem cell-like characteristics and extensive subculture capacity. Building on these cell resources, we further obtain cells adapted in the DKSFM media. The DKSFM-adapted cells have epidermal progenitor-like features and are responsive to extracellular calcium to undergo terminal differentiation. The approaches described in this paper could be useful tools to obtain epithelial cell lines for studying epidermal biology and diseases, such as developmental disorders, cancer and their underlying genetic risks.

## Supporting information

**S1 File. Supplemental materials and methods.**
(PDF)

**S2 File. Supplemental tables.**
(DOCX)

**S3 File. Supplemental figures.**
(PDF)

**S1 Raw images.**
(PDF)

## Acknowledgments

The authors would like to thank Dr. Alvaro Puga, University of Cincinnati, for critical reading, and Drs. Yuhang Zhang and Dorothy Supp, University of Cincinnati, for consultations on keratinocyte culture conditions.

## Author Contributions

**Formal analysis:** Jingjing Wang, Ying Xia.

**Methodology:** Jingjing Wang, Xiang Zhang, Ying Xia.

**Resources:** Maureen Mongan.

**Supervision:** Ying Xia.

**Writing – original draft:** Jingjing Wang.

**Writing – review & editing:** Ying Xia.

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
