## [Decision Letter · Decision Letter 0]

7 May 2021

PONE-D-21-10675

Isolation and long-term expansion of murine epidermal stem-like cells

PLOS ONE

Dear Dr. Xia,

Thank you for submitting your manuscript to PLOS ONE. After careful consideration, we feel that it has merit but does not fully meet PLOS ONE’s publication criteria as it currently stands. Therefore, we invite you to submit a revised version of the manuscript that addresses the points raised during the review process.

We look forward to receiving your revised manuscript.

Kind regards,

Majlinda Lako

Academic Editor

PLOS ONE

Journal Requirements:

2) As part of your revisions, please update your Methods to address the following items: (1) In the following section, "Experimental animals and cell culture," please clarify what is meant by "C57/BL6 mice were crossed." Do you mean "bred"?  and (2) specify method of euthanasia.

3) We note that you have included the phrase “data not shown” in your manuscript. Unfortunately, this does not meet our data sharing requirements. PLOS does not permit references to inaccessible data. We require that authors provide all relevant data within the paper, Supporting Information files, or in an acceptable, public repository. Please add a citation to support this phrase or upload the data that corresponds with these findings to a stable repository (such as Figshare or Dryad) and provide and URLs, DOIs, or accession numbers that may be used to access these data. Or, if the data are not a core part of the research being presented in your study, we ask that you remove the phrase that refers to these data.

4)  PLOS ONE now requires that authors provide the original uncropped and unadjusted images underlying all blot or gel results reported in a submission’s figures or Supporting Information files. This policy and the journal’s other requirements for blot/gel reporting and figure preparation are described in detail at https://journals.plos.org/plosone/s/figures#loc-blot-and-gel-reporting-requirements and https://journals.plos.org/plosone/s/figures#loc-preparing-figures-from-image-files. When you submit your revised manuscript, please ensure that your figures adhere fully to these guidelines and provide the original underlying images for all blot or gel data reported in your submission. See the following link for instructions on providing the original image data: https://journals.plos.org/plosone/s/figures#loc-original-images-for-blots-and-gels.

Reviewers' comments:

Reviewer's Responses to Questions

**Comments to the Author**

1. Is the manuscript technically sound, and do the data support the conclusions?

Reviewer #1: Partly

2. Has the statistical analysis been performed appropriately and rigorously? 

Reviewer #1: Yes

3. Have the authors made all data underlying the findings in their manuscript fully available?

Reviewer #1: Yes

4. Is the manuscript presented in an intelligible fashion and written in standard English?

Reviewer #1: Yes

5. Review Comments to the Author

Reviewer #1: The manuscript describes an incremental development of protocols to culture murine primary keratinocytes. The experiments used to indicate that a variant of KFSM medium containing 0.06mM Ca2+ permits long term expansion of primary murine keratinocytes are robust and the data do support the idea of long term culture. Reporting conditions for long term culture alone are not novel since other publications such as Hager et al Journal of Investigative Dermatology, 1999,112(6):971-976, have described similar findings, however the authors of this current manuscript have extended the transcriptomic analysis of the keratinocytes subjected to their culture conditions.

That said, it is unusual for primary cells to undergo at least 70 passages so it would have been useful if the authors could have provided an explanation for this enhanced proliferative capacity. Their data imply a degree of stem cell like characteristics in the cell population they generate (eg CD34, CD200, expression) but they have not indicated why such stem cells have this proliferative capacity. It would be useful to measure telomere lengths, expression of mTert and activity of the murine telomerase holoenzyme complex in addition to markers of cell senescence such as expression of p16 INK4a. Moreover, they do not provide analyses of the karyotype at early and late passage numbers. These additional data could be supportive of a putative epidermal stem cell identity especially since the doubling time of the cells grown in low calcium KSFM seems to decrease substantially after passage 20 (figure 2C). This could of course imply the selection of a more highly proliferative cell population but it could also imply cell transformation

Another drawback of the data provided is that there seems to be no attempt to correlate transcriptomic identity of the cell population to passage number. Given that doubling times decrease, it may be assumed that the transcriptome of the bulk population does change so taking a “snapshot” of gene expression does not provide an accurate description of a potentially changeable phenotypic environment. This potential change with increasing passage number would need to be quantified before the method can be confirmed to produce larger numbers of murine keratinocytes that are useful for studying epidermal biology and disease as stated in the discussion section. If data from these studies can be provided, the findings may be of interest to the readership of PLOSone

6. PLOS authors have the option to publish the peer review history of their article (what does this mean?). If published, this will include your full peer review and any attached files.

Reviewer #1: No

---

## [Author Response · Author response to Decision Letter 0]

21 Jun 2021

, we provide point-by-point responses to the reviewer’s comments and outline changes made in adherence to Journal’s requirements: 

1. Their data imply a degree of stem cell like characteristics in the cell population they generate (eg CD34, CD200, expression) but they have not indicated why such stem cells have this proliferative capacity. It would be useful to measure telomere lengths, expression of mTert and activity of the murine telomerase holoenzyme complex in addition to markers of cell senescence such as expression of p16 INK4a.

Authors’ responses: The keratinocyte lines reported here displayed some stem cell characteristics, including unlimited self-renew and prolonged in vitro culture capabilities, in addition to the expression of stem cell genes. Our new experiments, as suggested by the reviewer, showed that neither p16INK4a expression nor telomere length were significantly altered in cells at low and high passages. suggesting long-term stability of these cells. 

2. They do not provide analyses of the karyotype at early and late passage numbers.

Authors’ responses: This was an excellent point we had omitted previously. Karyotyping showed that even the primary mouse keratinocytes (P0) had significantly increased chromosomes consistent with what was reported before (Hammiller, 2015). We detected, accordingly, tetraploid or hyperdiploid chromosomes in our keratinocyte lines in both the early and late passages. These observations suggest that the chromosomes, originally unstable in primary keratinocytes, became relatively stable in the cell lines. 

3. Another drawback of the data provided is that there seems to be no attempt to correlate transcriptomic identity of the cell population to passage number. Given that doubling times decrease, it may be assumed that the transcriptome of the bulk population does change so taking a “snapshot” of gene expression does not provide an accurate description of a potentially changeable phenotypic environment.

Authors’ responses: With the additional data on telomere, p16 INK4a expression and chromosomal numbers, and the fact that the doubling time became stable in cells at “high (>30) passages”, we reason that the high passaging cell lines maintain stable molecular signatures, and hence, gene expression.

---

## [Decision Letter · Decision Letter 1]

2 Jul 2021

Isolation and long-term expansion of murine epidermal stem-like cells

PONE-D-21-10675R1

Dear Dr. Xia,

We’re pleased to inform you that your manuscript has been judged scientifically suitable for publication and will be formally accepted for publication once it meets all outstanding technical requirements.

Kind regards,

Majlinda Lako

Academic Editor

PLOS ONE

Reviewers' comments:

Reviewer's Responses to Questions

**Comments to the Author**

1. If the authors have adequately addressed your comments raised in a previous round of review and you feel that this manuscript is now acceptable for publication, you may indicate that here to bypass the “Comments to the Author” section, enter your conflict of interest statement in the “Confidential to Editor” section, and submit your "Accept" recommendation.

Reviewer #1: All comments have been addressed

2. Is the manuscript technically sound, and do the data support the conclusions?

Reviewer #1: Yes

3. Has the statistical analysis been performed appropriately and rigorously? 

Reviewer #1: Yes

4. Have the authors made all data underlying the findings in their manuscript fully available?

Reviewer #1: Yes

5. Is the manuscript presented in an intelligible fashion and written in standard English?

Reviewer #1: Yes

6. Review Comments to the Author

Reviewer #1: (No Response)

7. PLOS authors have the option to publish the peer review history of their article (what does this mean?). If published, this will include your full peer review and any attached files.

Reviewer #1: No

---

## [Editor Report · Acceptance letter]

7 Jul 2021

PONE-D-21-10675R1 

Isolation and long-term expansion of murine epidermal stem-like cells 

Dear Dr. Xia:

I'm pleased to inform you that your manuscript has been deemed suitable for publication in PLOS ONE. Congratulations! Your manuscript is now with our production department. 

Kind regards, 

on behalf of

Dr. Majlinda Lako 

Academic Editor

PLOS ONE